# Trans cohort metabolic reprogramming towards glutaminolysis in long-term successfully treated HIV-infection

Flora Mikaeloff[1], Sara Svensson Akusjärvi[1], George Mondinde Ikomey[2,3], Shuba Krishnan[1], Maike Sperk[1], Soham Gupta[1], Gustavo Daniel Vega Magdaleno[4], Alejandra Escós[1], Emilia Lyonga[2,3], Marie Claire Okomo[2,3], Claude Tayou Tagne[3], Hemalatha Babu[5,6], Christian L. Lorson[7,8], Ákos Végvári[9], Akhil C. Banerjea[10], Julianna Kele[11], Luke Elizabeth Hanna[5], Kamal Singh[7,8], João Pedro de Magalhães[4], Rui Benfeitas[12] & Ujjwal Neogi[1,13✉]

Despite successful combination antiretroviral therapy (cART), persistent low-grade immune activation together with inflammation and toxic antiretroviral drugs can lead to long-lasting metabolic flexibility and adaptation in people living with HIV (PLWH). Our study investigated alterations in the plasma metabolic profiles by comparing PLWH on long-term cART(>5 years) and matched HIV-negative controls (HC) in two cohorts from low- and middle-income countries (LMIC), Cameroon, and India, respectively, to understand the system-level dys-regulation in HIV-infection. Using untargeted and targeted LC-MS/MS-based metabolic profiling and applying advanced system biology methods, an altered amino acid metabolism, more specifically to glutaminolysis in PLWH than HC were reported. A significantly lower level of neurosteroids was observed in both cohorts and could potentiate neurological impairments in PLWH. Further, modulation of cellular glutaminolysis promoted increased cell death and latency reversal in pre-monocytic HIV-1 latent cell model U1, which may be essential for the clearance of the inducible reservoir in HIV-integrated cells.

[1] The Systems Virology Lab, Division of Clinical Microbiology, Department of Laboratory Medicine, Karolinska Institute, ANA Futura, Campus Flemingsberg, Stockholm, Sweden. [2] Center for the Study and Control of Communicable Diseases (CSCCD), Faculty of Medicine and Biomedical Sciences, University of Yaoundé 1, P.O. Box. 8445 Yaoundé, Cameroon. [3] Department of Microbiology, Haematology, Parasitology and Infectious Disease, Faculty of Medicine and Biomedical Sciences, University of Yaoundé 1, Yaoundé, Cameroon. [4] Integrative Genomics of Ageing Group, Institute of Life Course and Medical Sciences, University of Liverpool, Liverpool, United Kingdom. [5] Department of HIV/AIDS, National Institute for Research in Tuberculosis, ICMR, Chennai 600031, India. [6] Division of Microbiology and Immunology, Yerkes National Primate Research Center, Emory Vaccine Center, Emory University, Atlanta, GA 30329, USA. [7] Christopher S. Bond Life Sciences Center, University of Missouri, Columbia, MO 65211, USA. [8] Department of Veterinary Pathobiology, University of Missouri, Columbia, MO 65211, USA. [9] Division of Chemistry I, Department of Medical Biochemistry and Biophysics, Karolinska Institutet, Stockholm, Sweden. [10] National Institute of Immunology, Aruna Asaf Ali Marg, New Delhi, India. [11] Department of Physiology and Pharmacology, Neurovascular Biology and Health, Karolinska Institutet, 171 77 Stockholm, Sweden. [12] National Bioinformatics Infrastructure Sweden (NBIS), Science for Life Laboratory, Department of Biochemistry and Biophysics, Stockholm University, S-10691 Stockholm, Sweden. [13] Manipal Institute of Virology (MIV), Manipal Academy of Higher Education, Manipal, Karnataka, India. ✉email: ujjwal.neogi@ki.se

Combination antiretroviral therapy (cART) can effectively block replication of human immunodeficiency virus type-1 (HIV). However, persistent low-grade immune activation together with inflammation, despite successful long-term treatment, and toxic antiretroviral drugs can lead to long-lasting metabolic flexibility and adaptation in people living with HIV (PLWH)[1–3]. Metabolic alterations have earlier been reported in HIV infection in treated and non-treated PLWH that includes altered amino acid and fatty acid metabolism[1,4–6]. Furthermore, the metabolic adaptations associated with HIV infection are highly representative of immune dysregulation and inflammation related to accelerated aging[7,8]. However, there is a dearth of evidence about metabolic dysregulation due to successful long-term treatment in PLWH.

HIV regulates two essential immuno-metabolic pathways in the cell, namely glycolysis, and glutaminolysis, to sustain the availability of biomolecules needed for viral replication in a cell-type dependent manner[9]. The immune system is also adversely affected by HIV persistence and cell-to-cell spread that permits viral replication, despite on cART[10]. This can further trigger transient or persistent metabolic changes that drive immune-senescence and accelerated aging in PLWH. Moreover, studies have shown altered glutaminolysis with high plasma glutamate levels in PLWH and that the modified glutaminolysis is responsible for the late immune recovery following cART[4].

In our recent untargeted metabolomics study on the COCOMO cohort from Denmark, we reported alterations in the amino acid (AA) metabolism as a central characteristic of PLWH with a median of 13 years of therapy. This alteration was also more prominent in PLWH with metabolic syndrome (MetS)[3]. However, reproducibility of metabolomic studies is challenging due to variability in metabolites linked with environmental factors and population-based heterogeneity in diet, gut microbiome, and lifestyle choices such as smoking or alcohol that have crucial impacts on individual metabolite composition and concentration[11–13]. Finally, pre-analytical sampling errors, methodological errors, and informatics can influence the overall outcome and lead to cohort-specific effects.

This study investigates alterations in plasma metabolic profiles by comparing PLWH on long-term cART and matched HIV-negative controls (HC) in two cohorts from low- and middle-income countries (LMIC), Cameroon and India, respectively, using untargeted metabolomics to understand the system-level alterations during prolonged therapy in HIV-infection. We performed advanced network-based systems biology approaches and machine learning algorithms to identify the commonality between the cohorts associated with the cART in PLWH. We also used targeted metabolomics in a larger sample size of treatment naïve and experienced PLWH and HC from Cameroon and India to further validate our findings. Finally, to identify the metabolic state of the lymphocytic and promonocytic HIV-1 latent cell models, we performed in vitro studies by modulating the critical metabolic pathways identified in PLWH on cART in the presence and absence of the cART regimens. Our study provides a comprehensive metabolic profile of PLWH on cART, while HIV latency in cellular models can shed light on the metabolic reprogramming in long-term successfully treated HIV infection and its potential role in accelerated aging in PLWH.

## Results
### Clinical characteristics
We used a cohort of PLWH in Cameroon on cART ($n = 24$, with a median duration of treatment of 11 years [IQR 8.00–13.25]) and HIV-negative controls (HC, $n = 24$) (Table 1). All the patients were on a TDF/3TC/EFV-based regimen, non-smokers, and meat-eater. The only clinical parameter that achieved a statistically significant difference ($p = 0.007$) between HC and cART patients was exercise.

### Impaired amino acid metabolism in PLWH on cART in the Cameroon cohort
Using plasma untargeted metabolomics, a total of 841 metabolites were detected, of which 46% (390/841) were lipids, 22% (188/841) were amino acids (AA), and 17% (143/841) were xenobiotics (Fig. 1a). A low percentage of the metabolites were shown to be associated with diet, genetics, microbiome, lifestyle, and time of sampling as reported[11] (Supplementary Fig. 1). Out of all the detected metabolites, 122 metabolites differed significantly between PLWH on cART and HC (Mann–Whitney U-test, $p < 0.05$), of which 48% belonged to lipids (59/122) and 20% to AA (24/122). After correcting for multiple comparisons, 42 metabolites were statistically significant between PLWH on cART and HC (False discovery rate, FDR < 0.1) (Fig. 1a). Dimensionality reduction using these 42 metabolites showed an apparent clustering between HC and cART patients (Fig. 1b). Among the 42 metabolites, 45% (19/42) were less abundant in PLWH on cART than HC. To identify mechanisms associated with HIV and cART, the differentially abundant metabolites having a Human Metabolome Database (HMDB) annotation (herein Mann–Whitney U-test, $p < 0.05$) were submitted to metabolite set enrichment analysis (MSEA) using Ingenuity Pathway Analysis (IPA). Based on the IPA (Z-score > 2, FDR < 0.05), the top identified pathways were concentration of lipids ($n = 15$), synthesis of lipids ($n = 13$), and production of reactive oxygen species (ROS) ($n = 12$) (Fig. 1c). The commonality of these pathways was the presence of glutamate while other AA such as arginine, cysteine, and methionine were present in at least one pathway. These results showed a shift in lipid biosynthesis, immune activation of blood cells, and altered oxidative stress [production of reactive oxygen species (ROS) and hydrogen peroxide]. Metabolites within lipid metabolism exhibited the most significant difference in PLWH on cART compared to HC (FDR < 0.1) (Fig. 1d). In this cluster, the changes appeared heterogeneous with a higher abundance of seven metabolites and a lower abundance of 12 metabolites in PLWH. Furthermore, AA, energy, and nucleotide metabolism were highly interlinked and upregulated in PLWH compared to HC. To complement the MSEA results, a balanced Random Forest (RF) algorithm was trained to predict HC and cART status of each metabolite based on Metabolon terms. Then, a permutation feature importance was applied to this machine learning technique. This analysis identified AA followed by lipids as the top pathways (Fig. 1e) thereby confirming the importance of AA during cART in PLWH.

### Altered neurosteroids as a common factor for the two cohorts from LMIC
To identify common biomarkers associated with HIV status and the impact of cART and strengthen our study, we compared the data from our Cameroon cohort with untargeted metabolomics analysis from an Indian cohort. The study design of the Indian cohort was similar using advanced bioinformatics and statistical analysis. Initially, we took the overlapping result of a linear classification model (PLS-DA), a machine learning model, RF, and Mann–Whitney U-test, performed in the two cohorts separately. The two separated RF models had a predictive accuracy of 97.9% in Cameroon and 100% in the Indian cohort (Supplementary Fig. 2) after 10-fold cross-validation. Overlap between the metabolites in each cohort is presented in Supplementary Fig. 3. Then, we identified 14 (Mann–Whitney), nine (PLS-DA), and six (RF) biomarkers differing between HC and ART in both Cameroon and Indian cohorts. The overlap between all three methodologies identified six common metabolites to all

**Table 1 Clinical and demographic information.**

|  | Control | PLWH on cART | *p*-value |
|---|---|---|---|
| **Number** | 24 | 24 |  |
| **Age in years, mean (95% CI)** | 48.00 (45.59-50.41) | 47.54 (42.54-52.5) | 0.86* |
| **Gender, Female, *n* (%)** | 12 (50) | 12 (50) | 1** |
| **CD4 count, median (IQR)** | - | 753.5 (540.5-827.8) | - |
| **CD8 count, median (IQR)** | - | 754.0 (483.2-955.0) | - |
| **Viral load, <40 copies/mL, *n* (%)** | - | 24 (100) | - |
| **Duration of the current regime in years, median (IQR)** | - | 11.00 (8.00-13.25) | - |
| **Alcohol consumption, *n* (%)** | 15 (62.50) | 17 (70.83) | 0.75** |
| **Exercise, *n* (%)** | 4 (16.67) | 14 (58.33) | 0.007** |
| **BMI, mean (95%)** | 27.88 (26.47-29.29) | 26.66 (23.90-29.41) | 0.42* |

*Student *t*-test, **Chi-square test.

methods (Fig. 2a). After removing the antiviral drug efavirenz, five metabolites were overlapping between the two cohorts: 5α–androstan–3α,17β – diol monosulfate, androsterone sulfate, epiandrosterone sulfate, metabolonic lactone sulfate, and methionine sulfone (Fig. 2b). All the identified metabolites exhibited a similar trend between HC and cART in both cohorts thereby confirming their relevance. Furthermore, after correcting for confounders i.e. exercise using multivariate linear regression, all metabolites were statistically significant (Supplementary Fig. 4). To further investigate the interactions of the metabolites related to HIV status, we performed a metabolite co-abundance network analysis in the Cameroon cohort based on significant positive pairwise correlation (Spearman, FDR < 0.05) and used the Indian cohort for validation of the results. The network contained six communities found using Leiden algorithm (Fig. 2c). The most central community (community 5, with the highest mean degree) showed 110 metabolites where the majority were lipids (85%), showing potential lipid dysregulation in cART (Fig. 2d). Interestingly, four neurosteroids out of the five potential biomarkers; 5α–androstan–3α,17β – diol monosulfate, androsterone sulfate, epiandrosterone sulfate, and metabolonic lactone sulfate, were in the same community (community 4) containing a total 123 metabolites. Metabolites from this community were less abundant in PLWH on cART compared to HC while community 6, containing glutamate, showed higher abundance. To validate the robustness of our complement biomarkers discovery approach, we used the biomarkers and their first neighbors from the co-expression network to separate HC and PLWH on cART in the Cameroon cohort. Based on hierarchical and consensus clustering, segregation of HC and PLWH on cART was observed in the Cameroon cohort (Fig. 2e, Supplementary Fig. 5). As an independent validation, we performed a similar clustering in the Indian cohort (Fig. 2e, Supplementary Fig. 5) using the same metabolites set and found an even better separation between HC and PLWH on cART than in the Cameroon cohort. Therefore, our data identified a set of correlated biomarkers, mainly neurosteroids, that were associated with PLWH on cART in two independent cohorts that could be linked to potential neurological impairments.

**Targeted metabolomics in a larger cohort identified altered amino acid metabolism in two independent cohorts with differential mechanisms.** To validate the importance of AA (as observed in Fig. 1), we performed targeted metabolomics for AA in 90 PLWH on cART, 78 HC, and 45 untreated HIV-infected patients with viremia from Cameroon ($n = 123$) and India ($n = 90$). Even though the samples were run together, there was a clear cohort effect (Supplementary Fig. 6). Therefore, the subsequent analyses were performed on the cohorts separately. Eight AA were altered in the Cameroon cohort and 13 AA were altered

in the Indian cohort between PLWH on cART and HC. Among these, 6 AA were overlapping between both the cohorts (Fig. 3a, Supplementary Fig. 7), of which five essential AA including methionine, phenylalanine, threonine, valine, and tryptophan were significantly lower (FDR < 0.1) in PLWH on cART (Fig. 3b–f). Only glutamate was significantly higher in PLWH on cART compared to HC in both cohorts. Interestingly, glutamate was significantly higher in untreated HIV-infected PLWH than HC in the Indian cohort but not in the Cameroon cohort (Fig. 3g). Taken together these results indicate the importance of glutamate and AA in PLWH on cART. To assess the size of the difference over significance, the effect size calculation was performed based on the sample mean difference (ART-HC) using Glass delta ($D$). Overall, similar direction of the effect was observed in both cohorts (Fig. 3h–i). Glutamate tests were medium ($D = 0.62$) and small ($D = 0.25$) in Cameroon and Indian cohorts, respectively, and showed a greater mean in ART compared to HC. The largest effect sizes were observed in the Indian cohort on tryptophan ($D = -1.07$), serine ($D = -0.84$), and methionine ($D = -0.81$) (Fig. 3i). Overall, these results indicate alterations in glutaminolysis as a common phenomenon in long-term treated individuals and that glutamate, glutamine, and GABA plays an important role in metabolic reprogramming.

**Role of altered glycolysis and glutaminolysis in HIV latency cell models.** We observed alterations in the AA metabolism in PLWH on cART. Therefore, we aimed to study the effect of glycolysis and glutaminolysis during HIV latency, both of which are important for supplementing and utilization of AA and energy production by the TCA cycle (Fig. 3j). During cART, HIV persists in latent reservoirs both from monocytic and lymphocytic lineages. Earlier studies have shown how metabolic regulation may be a key factor regulating HIV infection where alterations in glycolysis have a large effect[14,15]. To characterize alterations in cellular metabolism during the steady-state of latent HIV infection, we performed quantitative proteomic analysis using LC-MS/MS in promonocytic latent cell model U1, and lymphocytic latent cell model J-lat 10.6 together with their respective uninfected parental cell lines, U937 and Jurkat. The steady-state modifications in the cell lines upon entering HIV latency showed significant alterations in the biosynthesis of AA and TCA cycle (Fig. 4a). First, we used 6-diazo-5-oxo-L-norleucine (DON) and 2-deoxy-D-glucose (2-DG) to block glutaminolysis and glycolysis, respectively, followed by a latency reversal agent prostratin to understand the roles of these pathways during latency reversal in J-Lat 10.6 and U1 cell lines. Both latency cell models, J-Lat 10.6 and U1 exhibited decreased viability when treated with 2-DG or DON compared to Jurkat and U937 cells (Fig. 4b). As expected, prostratin, a protein kinase C agonist, activated the latent reservoir in both cell lines (Fig. 4c, d). Interestingly, blocking

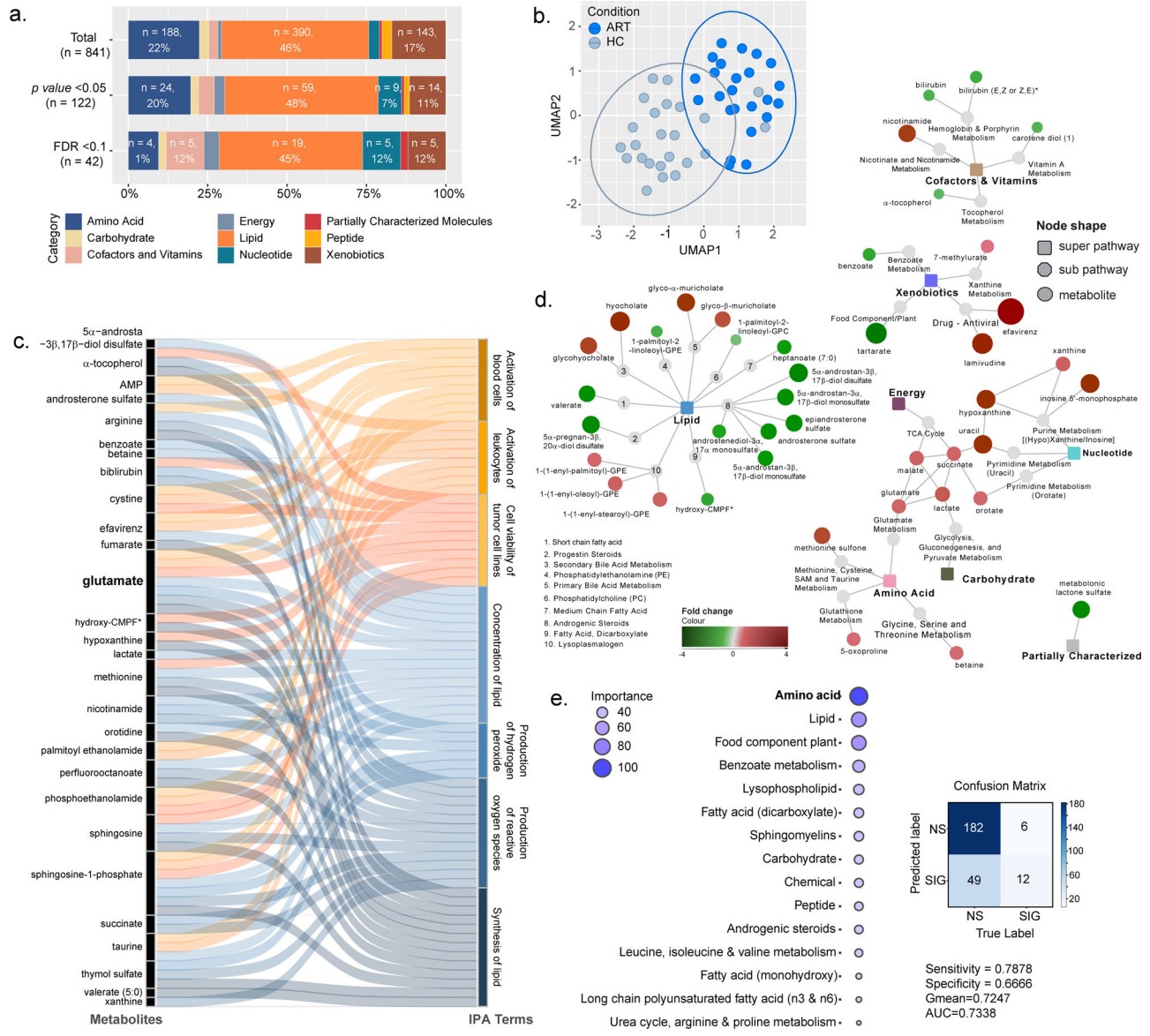

**Fig. 1 Untargeted metabolomics in the Cameroon cohort highlight substantial metabolic alterations in cART compared with HC. a** Bar plots representing the proportion of super pathways and number of associated metabolites in total detected metabolites ($n = 841$), metabolites with differential abundance between HC and PLWH on cART with $p < 0.02$ (Mann–Whitney U-Test, $n = 122$) and FDR < 0.1 (Mann–Whitney U-test, $n = 42$). **b** UMAP visualization of 48 samples using metabolites differing between HC and PLWH on cART (Mann–Whitney U-test, FDR < 0.1, $n = 42$). Samples are colored by condition (light blue = HC; dark blue=cART). **c** Sankey Plot illustrating the most important contribution to the flow of glutamate-associated pathways together with metabolites that are altered in cART patients. **d** Network of the metabolites significantly differing between HC and cART (Mann–Whitney U-test, FDR < 0.1, $n = 42$). Colored rectangular nodes represent super pathways, gray circles subpathways, and colored circles single metabolites. The color gradient was applied depending on log2FC for each metabolite from green (decreased in cART) to red (increased in cART). The size of the bubble is proportional to log2FC. Edges connect each metabolite to its respective subpathway and each subpathway to its respective super pathway. **e** Bubble plot showing the importance of Metabolon pathways in the prediction of metabolite association with cART status and the associated confusion matrix and classifier metrics. Terms represented at the top of the figure are the most important for prediction. (RF, estimators: 500, class weight: balanced).

glutaminolysis using DON alone activated the latent reservoir in U1 cells (Fig. 4d) but not in J-Lat 10.6 cells (Fig. 4c). To further explore the alterations concomitant with blocking glutaminolysis, we performed quantitative proteomic analysis using LC-MS/MS in U1 and U937 cells following treatment with DON, prostratin, and DON + prostratin. Because differences in the steady-state protein levels are observed between the two cell lines, we corrected the effect of U937 in U1 cells and performed differential abundance analysis and protein set enrichment analysis (Fig. 4e). The addition of prostratin alone in U1 cells showed alterations in KEGG metabolism pathways such as carbon metabolism, TCA cycle and AA metabolism ($p < 0.1$, Supplementary Fig. 8), which could be linked to latent HIV reservoir activation. Exposure to DON showed dysregulation of several metabolic pathways including glycolysis/gluconeogenesis, TCA cycle, sulfur metabolism, valine, leucine, isoleucine degradation and oxidative phosphorylation (OXPHOS). Several proteins of OXPHOS complexes I, III and IV, displayed lower abundance in U1 + DON compared to U1 control cells (Fig. 4f). To confirm this, we performed western blot using human OXPHOS antibody cocktail that detects the components of complex-V (ATP5a), complex-III (UQCRC2), complex-II (SDHB), complex-IV (COX II), and

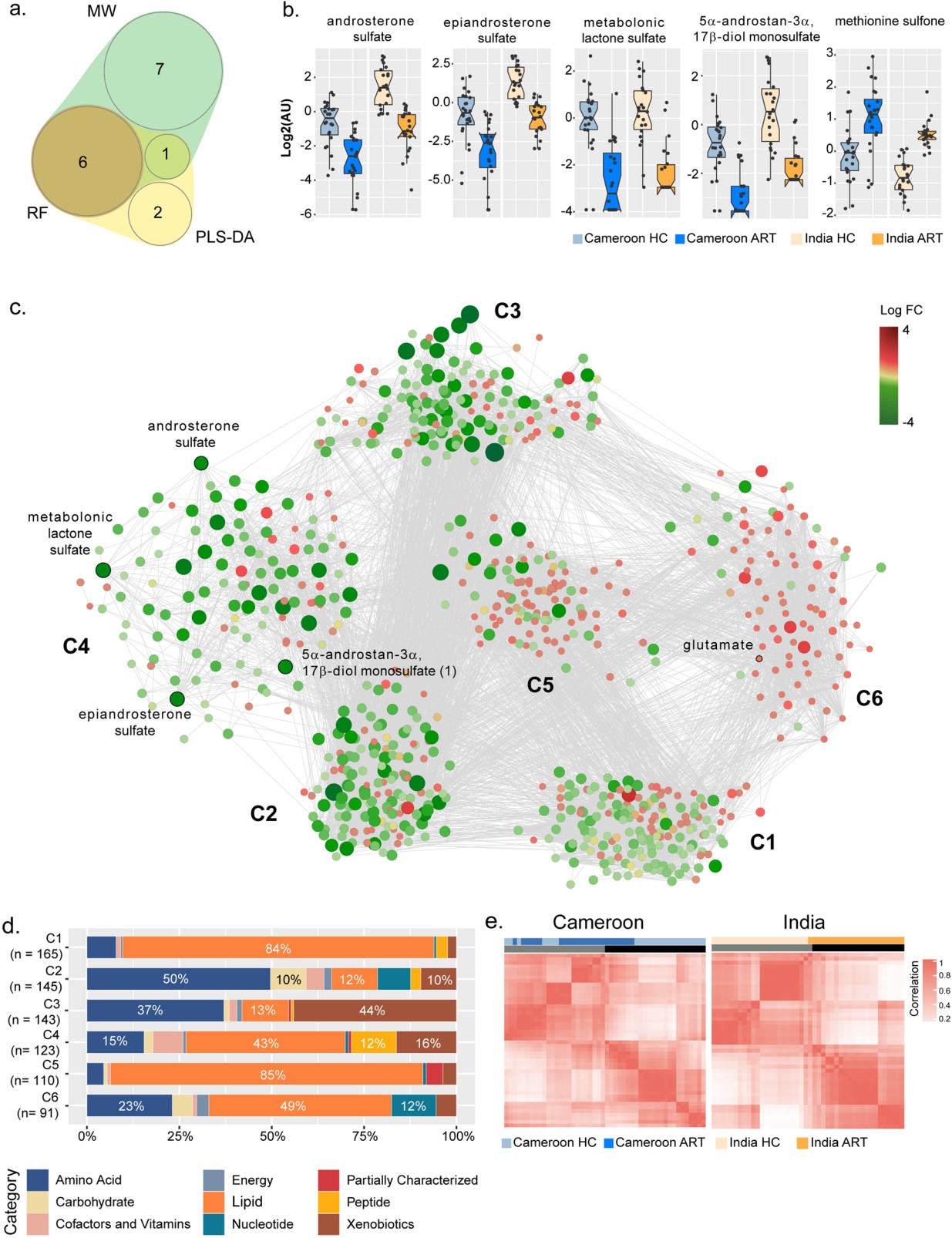

complex-I (NDUFB8). Since we could not differentiate complex-I and -IV in all our western blots, we considered them together in our analysis. While we did not observe any statistically significant differences in the protein levels of the OXPHOS complexes, visible decrease in expression of complexes-III, -II, -I, and -IV but not complex-V were observed upon DON treatment both in the

presence and absence of prostratin (Fig. 4h). Overall, our experimental data correlated with the proteomics data and is suggestive of suppression of OXPHOS upon blocking glutaminolysis in HIV latently infected promonocytic cells. Since changes in OXPHOS are often linked to unbalanced redox homeostasis, we measured the total cellular ROS levels in U1 cells following

**Fig. 2 Identification of biomarkers associated with HIV status and impact of cART compared to HC. a** A 4-dimensional, quasi-proportional Venn diagram showing the number of overlapping metabolites ($n = 6$) differing HC/cART from three methodologies (Mann–Whitney $U$-test, RF, and PLS-DA) in Indian and Cameroon cohort. Analysis was performed separately in Indian and Cameroon cohorts. **b** Box plots of significant biomarkers shared by Indian and Cameroon patients: androsterone sulfate, epiandrosterone sulfate, metabolomic lactone sulfate, 5α–androstan–3α,17β-diol monosulfate, and methionine sulfone. In all the comparisons, FDR < 0.001 Dots represent individual values and the line represents the median. **c** Global association analysis network and identified communities. Potential biomarkers and glutamate are indicated. **d** Bar plots representing proportion of super pathways and number of associated metabolites in communities ($n\_c1 = 165$, $n\_c2 = 145$, $n\_c3 = 143$, $n\_c4 = 123$, $n\_c5 = 110$, $n\_c6 = 91$). **e** Consensus matrices of potential biomarkers and first neighbors in HC and cART. Data were log-transformed and z-score transformed. Cameroon HC (light blue); Cameroon cART (dark blue); India HC (light yellow); and India cART (dark yellow).

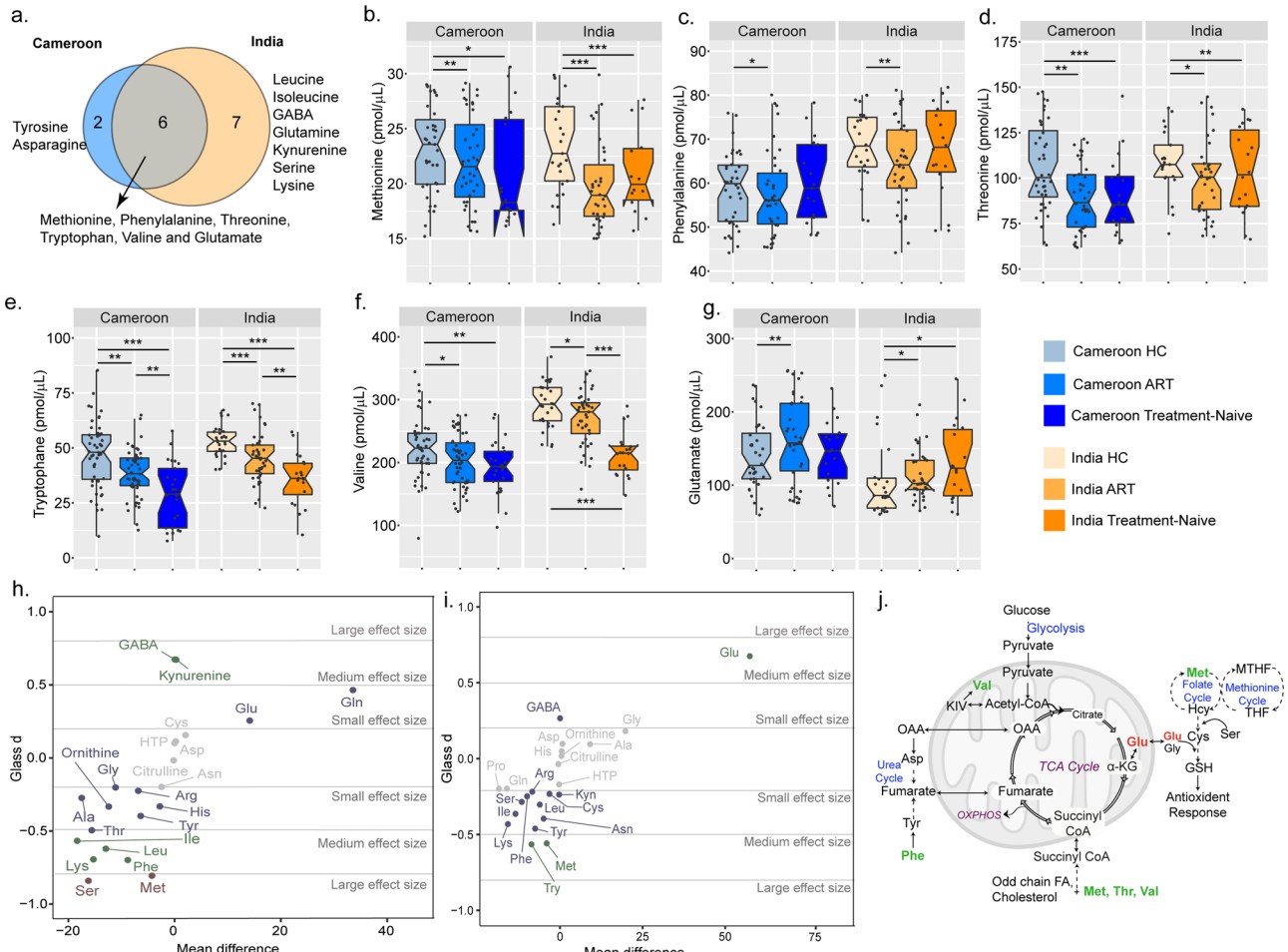

**Fig. 3 Targeted amino acid in the larger HIV-1 cohorts from Cameroon and India. a** Venn diagram representing the overlap of AA significantly differing in HC compared to cART between Cameroon and Indian cohort. **b**–**g** Box plots showing the abundance of six significant AA between HC and cART (Mann–Whitney $U$-test, FDR < 0.1) (*FDR < 0.1, **FDR < 0.05, ***FDR < 0.01): methionine (**b**), phenylalanine (**c**), threonine (**d**), tryptophan (**e**), valine (**f**), and glutamate (**g**) in HC, and PLWH on cART and treatment-naive patients from Cameroon and Indian cohorts. Cameroon HC (light blue) $n = 50$; Cameroon cART (blue) $n = 50$; Cameroon Treatment-Naïve (dark blue) $n = 25$; India HC (light yellow) $n = 30$; India cART (light orange) $n = 41$; India Treatment-naïve (orange) $n = 20$. **h** and **i** Scatter plot of AA mean differences by effect size (Glass delta ($D$)) in Indian (**h**) and Cameroon (**i**) cohorts. Dots are colored based on effect size (red = large, green = medium, blue = small). **j** Schematic representation of the altered AA linked with the key metabolic pathways.

treatment with prostratin, DON, and DON + prostratin (Fig. 4i). We observed a decrease in cellular ROS levels in the presence of prostratin compared to the control. However, inhibition of glutaminolysis by DON did not alter ROS levels compared to control cells ($p > 0.05$, ns). Conclusively, it can be postulated that inhibition of glutaminolysis can compensate for the effect of HIV latency by altering oxidative stress. Therefore, ROS may not be the obligate driver of metabolic reprogramming in the cell models of HIV latency.

**2-DG and DON modulates intracellular metabolite levels independent of ART regimens in U1 cells.** Thereafter, we sought to evaluate the effect of 2-DG and DON during latency reversal in the presence of cART regimens; tenofovir disoproxil fumarate (TDF) + lamivudine (3TC) + efavirenz (EFV) (TDF + 3TC + EFV), and zidovudine (AZT) + 3TC + EFV (AZT + 3TC + EFV) prevalently used in low-income countries. Cellular cytotoxicity of cART regimen was evaluated using Alamar blue assay (Supplementary Fig. 9). In our promonocytic latency model, U1,

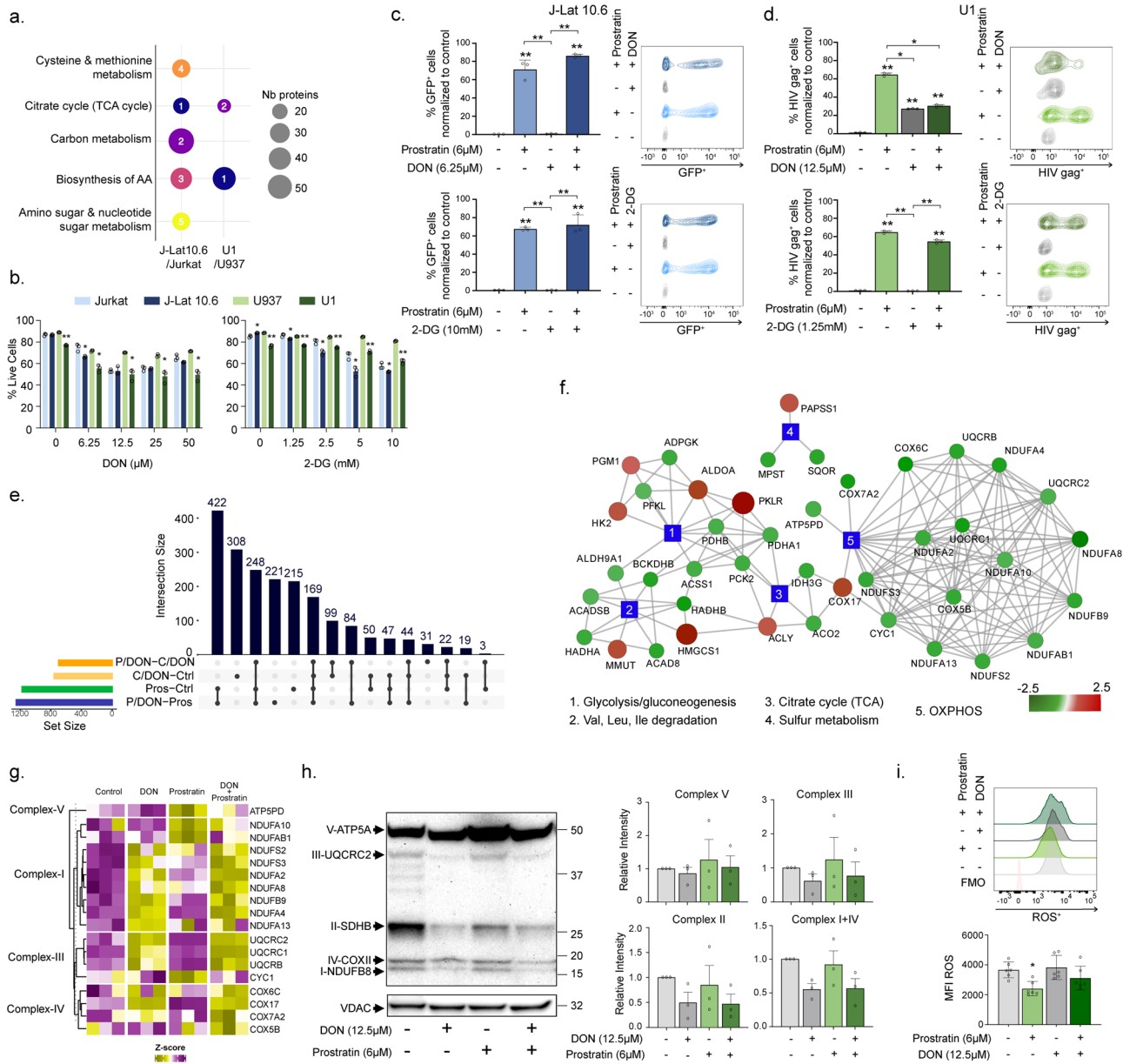

**Fig. 4 Effect on cellular metabolism in latency cell models during latency reversal. a** Steady-state metabolic alterations in the HIV latency cell models, J-Lat 10.6 (dark blue) and U1 (dark green), compared to Jurkat (light blue) and U937 (light green), respectively. MSEA using the KEGG Metabolism with FDR < 0.05 is shown as bubble plots. The size of the bubble represents the number of proteins and the number, the rank of the pathway based on FDR. **b** Viability of latency cell models J-Lat 10.6 and U1 compared to respective parental cell lines Jurkat and U937 during 48 h DON or 2-DG treatment measured using flow cytometry. **c, d** HIV latency activation using prostratin (6 μM) together with DON (6.25 μM) or 2-DG (10 mM) in J-Lat 10.6 cell line (**c**) and DON (12.5 μM) or 2-DG (1.25 mM) in U1 cell line (**d**). Data represented as bar graphs (mean ± SD) of three independent experiments. Flow cytometry contour plots are from a representative sample. **e** The upset plot of proteins with differential abundance between control vs. DON (C/DON-Ctrl, yellow), control vs. prostratin (Pros-Ctrl, green), prostratin vs. DON + prostratin (P/DON-Pros, blue), and DON + prostratin vs. DON (P/DON-C/DON, orange) in U1 cells corrected for U937 cells. Horizontal bars show the number of proteins found in each comparison. Vertical bars display intersects between comparisons as indicated in the matrix below the graph. **f** Network of the proteins differing significantly between control U1 and DON treated U1 (LIMMA, FDR < 0.1, n = 758). Blue colored rectangular nodes represent KEGG pathways and colored circles represent proteins. The color gradient was applied depending on log2FC for each metabolite from green (decreased in U1-DON) to red (increased in U1-DON). The size of the bubble is also proportional to log2FC. **g** Heatmap showing OXPHOS proteins levels in U1 treated with DON and prostratin. Proteins were selected based on comparison U1 vs U1-DON (LIMMA, FDR < 0.1) and their association with the OXPHOS KEGG pathway. Proteins were separated based on their complexes (from I to V). **h** Western blot analysis and quantification of OXPHOS proteins during DON treatment in U1 cells with a representative blot of three independent experiments (Fig. S10) are shown here. Quantification of western blot represented as bar graphs with mean ± SEM. **i** Measurement of ROS in latency cell model U1 during prostratin and DON treatments. Data represented as mean ± SD of three independent experiments All statistical analysis was performed using unpaired *t*-test or Mann–Whitney *U*-test (**p* < 0.05, and ***p* < 0.001).

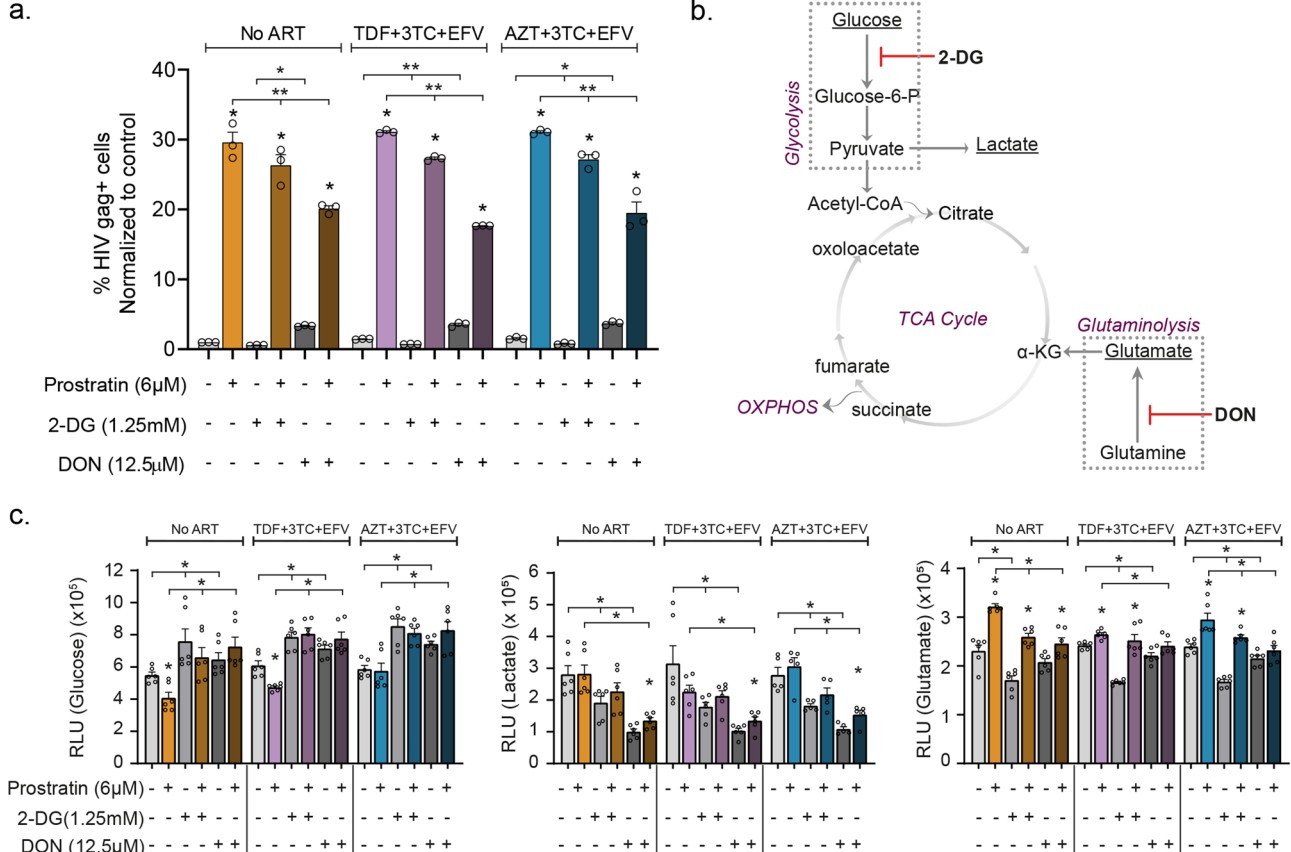

**Fig. 5 Effect of cART regimens on HIV activation during inhibition of glutaminolysis: effect of cART regimens (TDF + 3TC + EFV and AZT + 3TC + EFV) in combination with 2-DG and DON on HIV latency activation in U1 cells. a** Production of HIV gag during cART treatment in the presence of metabolic blockers 2-DG or DON. **b** Schematic showing the effect of 2-DG and DON on metabolic processes. **c** Effect of cART regimens on intracellular glucose, lactate, and glutamate levels when treated with 2-DG or DON during latency activation. All experiments were performed in three independent replicates. Statistical analysis was performed using the Mann–Whitney $U$-test (*$p < 0.05$, and **$p < 0.001$) and represented as mean ± SEM.

prostratin increased HIV gag protein expression independent of the cART regimens. However, both 2-DG and DON in combination with prostratin reduced the expression of gag (Fig. 5a). The increased gag expression by DON alone, as observed earlier, was detected independently of cART. Furthermore, to understand how 2-DG and DON modulate intracellular metabolite levels, we used metabolite measurement kits to measure glucose, lactate, and glutamate levels during latency reversal under cART pressure in U1 cells (Fig. 5b). Similar trends could be seen for the measured metabolites in the presence and absence of cART (Fig. 5c). Intracellular glucose levels decreased during prostratin treatment while 2-DG and DON increased glucose levels both with and without prostratin compared to control. Intracellular lactate levels were reduced during prostratin treatment both with and without the inhibitors compared to control. Interestingly, when treating U1 with prostratin and DON, lactate levels significantly increased compared to DON alone, independent of cART. Furthermore, prostratin increased intracellular glutamate levels both alone and with 2-DG or DON in absence of cART while under cART pressure the increase was only seen using prostratin or prostratin+ 2-DG alone. As 2-DG reduced glutamate levels, this increase could be caused by restitution of glutamate levels by prostratin.

## Discussion

Our study identified altered plasma AA profiles in two HIV cohorts of PLWH, from Cameroon and India, on successful cART. The common factor in this trans-cohort study was the lower level of essential AAs methionine, phenylalanine, threonine, valine, and tryptophan and elevated glutamate in PLWH on cART compared to HC. Significantly lower levels of neurosteroids like 5α–androstan−3α,17β–diol monosulfate, androsterone sulfate were observed in both cohorts. Modulation of cellular glutaminolysis increased cell death and latency reversal in promonocytic HIV latency cell model U1. Therefore, our study highlights the importance of altered glutaminolysis in PLWH that can be linked to accelerated neurocognitive aging and metabolic reprogramming in latently infected cells.

The metabolic network of AA is co-regulated and highly complex. Therefore, we conducted a trans-cohort metabolic profile analysis using advanced statistical analysis, machine learning algorithms, together with network analysis to identify metabolic features in PLWH. Our findings indicated an in-depth dysregulation of AA metabolism in long-term treated PLWH. Earlier studies on untreated and short-term treated PLWH (up to 36 months) on smaller cohorts from USA, Netherlands, and Spain have shown disruption of AA metabolism during infection[16–18]. In our study, we showed that AA dysregulation is a persistent feature even with >5 years of treatment. This result is in line with our recent large-scale metabolomics study from the Copenhagen Comorbidity in HIV infection (COCOMO) cohort with a median duration of 13 years treatment. In the COCOMO cohort, we observed altered AA metabolism in PLWH with and without metabolic syndrome (MetS) compared to the HC. Alterations in AA such as tryptophan, glutamine, glutamate, phenylalanine, arginine, aspartate, and threonine have been

closely linked to HIV and cART-induced metabolic complications as well as oxidative stress[19,20].

In our current study, we observed increased plasma levels of glutamate in PLWH on cART compared to HC in both cohorts, while it was only in the Indian cohort that treatment-naive PLWH had increased glutamate levels. Interestingly, in the Cameroon cohort treatment-naive PLWH had median glutamate levels like HC. One of the potential reasons could be that pre-therapy PLWH from India had a much lower CD4 count [median (IQR): 300.5 (213.2–527.0) cells/uL] compared to Cameroon cohort [median (IQR): 495.0 (382.0–558.0) cells/uL]. Possibly a severe depletion of CD4+ T-cells can influence the metabolic environment and cause detrimental effects. This hypothesis is further supported by our earlier studies showing PLWH with MetS have significantly lower nadir CD4+ T-cell count[3].

Glutamate displays remarkable metabolic versatility in several metabolic pathways including both AA synthesis as well as degradation. However, glutamate-mediated excitotoxicity is the primary contributor to age-related neurodegenerative disorders[21]. Elevated plasma glutamate in PLWH may cause loss of lymphocyte and macrophage function[22]. The primary source of extracellular glutamate in HIV infection includes the release of intercellular glutamate due to cell death, macrophages/microglia activation, and disrupted neurotransmitter clearance[23]. Metabolic tracer experiments demonstrated considerable changes in the glutamine metabolism with increased secretion of glutamine-derived glutamate from HIV-infected CD4+ T-cells[24]. An earlier study also reported that HIV infection enhances glutamate production in human monocyte-derived macrophages that may be an essential link to HIV-associated dementia[25]. In pre-clinical studies of EcoHIV Murine Model of HIV-associated neurocognitive disorders (HAND), the inhibition of the glutaminase (GLS) with DON or a DON prodrug JHU083 reversed the impaired cognitive function, indicating the role of the glutaminolysis in HAND[26,27]. In our study, we observed increased plasma glutamate in both the cohorts and low neuroactive steroids in PLWH with therapy compared to the HC. A recent study in PLWH with high and low depressive symptoms also reported a reduced level of neuroactive steroids in participants with high depressive symptoms[28] indicating that depression severity associated with lower levels of neuroactive steroids. Interestingly, neuroactive steroids have been shown to regulate glutamatergic neurotransmission (can bind to receptors for glutamate) as well as behavioral actions[29]. Taken together, it can be hypothesized that increased plasma glutamate and decreased neurosteroids in PLWH following successful therapy have the potential to develop neurocognitive impairment and depressive disorders that may need clinical intervention.

Metabolic reprogramming occurs to ensure energy availability and to elicit an appropriate immune response upon pathogen encounter. Susceptibility to HIV is partially regulated by the activation stage and the metabolic activity of a cell where elevated OXPHOS and glycolysis favors infection in lymphocytes[30,31]. Even as the main HIV reservoir is believed to reside in long-lived lymphocytic cells, latently infected monocytes and macrophages can persist over time and facilitate spread during cART. In this study, we show the effect of inhibition of glutaminolysis on latent reservoir in monocytic cells. On a metabolic level, this results in a reduction of proteins involved in OXPHOS while increasing proteins involved in glycolysis, proteinogenic branched-chain AA degradation (valine, leucine, and isoleucine), and TCA cycle. Previous studies have shown increased glutamine metabolism in latently infected cells[32–34]. Specifically, latently infected macrophages use glutaminolysis as a primary energy source in addition to fatty acid and glucose used by their uninfected counterparts[35]. Furthermore, macrophages carrying latent

HIV have a compromised TCA cycle that induces lipid accumulation and OXPHOS and enlarged mitochondrion[35]. In our study, inhibition of glutaminolysis was accompanied with altered metabolite levels, more specifically increased intracellular glucose and reduced glutamate and lactate levels irrespective of cART treatment. Uptake of glucose through Glut1 regulates susceptibility of HIV-1 and lymphocytes carrying latent HIV-1 have also been proposed to express OX40 together with Glut1[36,37]. Elevated glycolytic activity, Glut1 expression, and immune activation are also seen in people living with HIV and factors needed for virion production[38,39]. Herein, we did not see any effect on latency reversal when inhibiting glycolysis. Therefore, while glycolysis is an important factor for HIV-1 entry and replication, latency reversal is somewhat dependent on glutaminolysis. Furthermore, proteins involved in mitochondrial respiration were more abundant in latent monocytic U1 cells compared to parental U937 cell line indicating that OXPHOS proteins may play a role in HIV persistence. Therefore, a unique metabolic environment may be induced by the virus to maintain a transcriptionally inactive state. By inhibiting glutaminolysis, cellular metabolism can be altered and thereby force transcriptional activation of latent HIV-1.

The study has some limitations that merit comments. First, it is a cross-sectional study so the analysis was restricted to the association study, no causality can be inferred. Second, despite having the patients from two cohorts the number of samples was relatively small. Third, though we corrected multiple hypotheses, the high number of associations may have led to type I errors. Finally, the Cameroon cohort was a clinically highly selected cohort with long-term successful therapy that may not represent the general population. However, the biggest strength of our study is the use of two cohorts to validate the findings and identify the commonality between the cohorts associated with successful cART in PLWH.

In conclusion, our present study based on cohorts (India and Cameroon) indicated altered AA metabolism and more potentially a switch in glutaminolysis as the alternative pathway for energy production following long-term antiretroviral therapy. Altered glutaminolysis with long-term treatment and its association with metabolic syndrome[3], diminished immune recovery[4], and glutamate excitotoxicity mediated neurocognitive impairments can lead to increased co-morbidities and accelerated aging in PLWH with successful therapy. In addition, a decrease in neurosteroids causes major depressive syndrome[28] leading to diminished quality of life despite successful treatment. Our study also provided evidence displaying the cross-talk between glutaminolysis, TCA cycle, and OXPHOS in HIV-latent cell model being more specific to promonocytic U1 cells that potentially is linked with apoptosis as well as latency reversal. Increased knowledge about the co-regulation of interconnected metabolic pathways in the context of HIV infection and therapy can provide new targets for future therapeutic interventions both for improving metabolic health and other metabolic disorders in PLWH. It can also reveal a potential for clearing the latent reservoir by modulating the cellular metabolic pathways as a novel strategy for a functional cure.

## Materials and methods

**Study population**. This study included three groups of individuals in Cameroon: HIV-1 infected individuals on combination ART (cART, $n = 50$), untreated HIV-1 infected individuals with viremia (treatment-naive, $n = 25$), and HIV-negative individuals (HC, $n = 50$). The study groups are age- and BMI- matched with comparable gender proportions between males and females. Whole blood and plasma were collected from Yaounde University Teaching Hospital, Cameroon. Additionally, participants from an Indian cohort were included PLWH on cART ($n = 41$), treatment naïve ($n = 20$), and HC ($n = 30$), as reported earlier[2].

**Ethical clearance**. The Cameroon study was approved by the Cameroon National Ethics Committee for Human Research with Ethical clearance N$^O$2019/08-198-CE/CNERSH/SP. The Indian study was approved by the Institutional Ethics Committee of the National Institute for Research in Tuberculosis (NIRT IEC No: 2015023) and the Institutional Review Board of the Government Hospital for Thoracic Medicine (GHTM-27102015) Chennai, India. Ethical approval by Etik-prövningsmyndigheten (Sweden) was waived off because of the anonymized data (Dnr 2019-05086). Written informed consent was obtained from all study participants before inclusion and kept at the respective sites.

**Cell lines**. For this study, latency cell models J-Lat 10.6 (AIDS reagent program) and U1 (AIDS reagent program) were used together with their respective parental cell lines Jurkat (AIDS reagent program) and U937 (Kindly provided by Helena Jernberg Wiklund, Uppsala University). All the cells were maintained in RPMI-1640 (Sigma, USA) supplemented with 10% fetal bovine serum (Gibco, USA) and 20 units/mL penicillin and 20ug/mL streptomycin (Sigma, USA) under 5% $CO_2$ and 37 °C temperature.

**Untargeted and targeted metabolomics**. Plasma untargeted metabolomics was performed at Metabolon, Inc. (North Carolina, USA) on a selection of individuals from the Cameroon cohort including HC ($n = 24$) and cART ($n = 24$), as previously described[1,40]. The method is ISO 9001:2015 certified. A larger cohort targeted metabolomics for amino acids was performed using LC-MS/MS method with reference amino acids as control at the Swedish Metabolomics Centre (Umeå, Sweden). The detailed methodology is presented in the supplementary method. The patients' clinical and demographic data are presented in Supplementary Table 1. For detailed method and quality control please see supplementary method. Untargeted proteomics was performed at the Proteomics Biomedicum facility, Karolinska Institutet as described by us recently[41].

**Pathways and clustering analysis**. Pathway Analysis software (IPA) software (QIAGEN Inc., USA) and Metabolon terms were used in metabolomics data for pathway analysis. For proteomics, curated Metabolic KEGG Human 2019 libraries and enrichr function from gseapy python package were applied (version 0.10.3; https://github.com/zqfang/gseapy). Cutoff for pathways was set to FDR < 0,05. For association analysis, a significant positive correlations network (Spearman, FDR < 0.05) was built using python igraph (https://igraph.org/python/). Community analysis was done using Leiden algorithm and most central communities were found using mean degree[42]. The complete methodology was described before[41]. Cluster analysis was conducted using R package ConsensusClusterPlus with the following metrics [algo: hierarchical clustering, distance: spearman, pItem: 0.8, reps: 1000][43].

**Visualization**. Dimensionality reduction was performed using UMAP[44]. R package ggplot2[45], ggalluvial[46], nVennR[47], VennDiagram[48], UpsetR[49], and Complexheatmap[50] were used to generate figures. Cytoscape ver 3.6.1 was used for network representation[51].

**Cytotoxicity assays**. Toxicity of cART regimens (TDF + 3TC + EFV and AZT + 3TC + EFV) or 2-DG and DON were evaluated using AlmarBlue assay (Invitrogen) over the course of 5 days, and 48 h, respectively, according to the manufacturer's protocol.

**Flow cytometry**. Activation from latency was measured using GFP (J-Lat 10.6) or HIV-1 core antigen-FITC, KC57 (Beckman Coulter) staining (U1) complemented with Near-IR viability (Invitrogen). Cellular ROS was measured using CellROX$^{TM}$ Deep red Flow Cytometry Assay Kit (Invitrogen), according to the manufacturer's protocol. Experiments were run on Fortessa flow cytometer (BD Bioscience) and analyzed using FlowJo 10.6.2 (TreeStar Inc).

**Intracellular metabolite measurement**. Intracellular metabolites were measured using Glucose-Glo$^{TM}$ Assay, Lactate-Glo$^{TM}$ Assay, and Glutamate-Glo$^{TM}$ Assay (Promega) according to the manufacturer's protocol.

**Determination of mitochondrial OXPHOS**. U1 and U937 cells (seeding density: $10 \times 10^6$ cells/well) were either left untreated for 48 h or treated with DON (12.5 μM for 48 h), prostratin (untreated for 24 h followed by 6 μM prostratin for 24 h), or DON + Prostratin (12.5 μM DON for 48 h and 6 μM prostratin for last 24 h). At 48 h, activation of HIV was measured by intracellular staining of HIV-1 core antigen-FITC, KC57 as mentioned earlier. Cells were harvested, washed in PBS, and centrifuged. Cell pellets were processed for mitochondrial extraction using Mitochondria Isolation Kit for Cultured cell (Thermo Scientific) using a reagent-based method as per manufacturers guidelines, followed by measuring OXPHOS protein levels using the total OXPHOS Human WB antibody cocktail (Abcam) with mitochondrial loading control VDAC using the antibody VDAC clone B-6 (Santa Cruz Biotechnology). Relative protein quantification was performed using ImageLab version 6.0.1 (Bio-Rad Laboratories Inc), results

analyzed using Mann–Whitney $U$-test or unpaired $t$-test and visualized using GraphPad Prism 8.4.3 (significance $p < 0.05$). All laboratory experiments were performed in three independent replicates. Analysis was performed using unpaired $t$-test or Mann–Whitney $U$-test and visualized using GraphPad Prism 8.4.3 (significance $p < 0.05$). Uncropped and unedited blot images are included as Fig. S10.

**Statistics and reproducibility statement**. The Non-parametric Mann–Whitney $U$-test, Spearman correlations as well as linear regression were performed using R (https://www.r-project.org/). Machine learning models were built using R packages Boruta for feature selection[52], randomForest with 10-fold cross-validation, and python scikit-lean[53] and verified using confusion matrices and the Area under the Receiver Operator Curve (ROC). PLD-DA plot was made using ropls[54]. For proteomics, preprocessing was performed as described previously[41] and differential abundance analysis using R package LIMMA[55] using the following design: [~cell*$p$ + cell*don + cell:$p$:don]. False discovery rate (FDR) was applied for multiple comparisons correction. Effect sizes were calculated using the d_glass function from the R package effect size to compensate for relevant differences in the standard deviations.

**Reporting summary**. Further information on research design is available in the Nature Research Reporting Summary linked to this article.

## Data availability
The normalized metabolomics and proteomics data can be obtained from https://doi.org/10.6084/m9.figshare.14984415.v1. All the codes were available https://github.com/neogilab/Cameroon_India. The mass spectrometry proteomics data have been deposited to the ProteomeXchange Consortium (http://proteomecentral.proteomexchange.org) via the PRIDE partner repository[56] with the dataset identifier PXD028531.

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

## Acknowledgements

The study is supported by the Swedish Research Council (2017-01330, 2018-06156 and 2021-01756), Karolinska Institutet Stiftelser och Fonder (2020-1554), and Åke Wiberg Stiftelse grant (M18-0021). Swedish Physicians Against AIDS Foundation (FOb20170004) and Jeanssons Stiftelser (JS2016–0185) to UN. MS acknowledges the support received from the Swedish Physicians Against AIDS Foundation (FOa2019-0020). SG acknowledges support from the Swedish Research Council Establishment grant (2021-03035), the Center for Medical Innovation grant (CIMED-FoUI-093304), Karolinska Institutet Stiftelser och Fonder (2020-02153), and Åke Wiberg Stiftelse grant (M20-0220). We thank all the study subjects for their participation. Authors acknowledge support from the Proteomics Biomedicum; Karolinska Institute, Solna, for LC-MS/MS analysis. Swedish Metabolomics Centre, Umeå, Sweden is acknowledged for targeted metabolic profiling.

## Author contributions

Conceptualization and study designing: U.N., Clinical data, and biobank: G.M.I., E.L., M.C.O, C.T.T., H.B. L.E.H., Methodology: F.M., R.B., J.P.d.M., and U.N., Formal analysis (bioinformatics): F.M. and G.D.V.M. and R.B., Formal analysis (experimental): S.S.A., S.K., M.S., A.E., A.V., and S.G., Supervision: R.B., J.P.d.M., and U.N., Resources: U.N. Writing (original draft): F.M., S.S.A, S.K., M.S., and U.N. Writing (review and editing): R.B., G.D.V.M., A.C.B, J.K., K.S., C.L.L., and J.P.d.M., Visualization: F.M., S.S.A, S.K. and U.N., Project administration: G.M.I., L.E.H., and U.N. Funding acquisition: U.N. All authors discussed the results, commented, and approved the final version of the manuscript.

## Funding

## Competing interests

C.L.L. is the co-founder and chief scientific officer for Shift Pharmaceuticals. The remaining authors declare no competing interests.
