## [Peer Review File · Communications Biology]

Reviewers' comments:

Reviewer #1 (Remarks to the Author):

This study investigates the metabolic alterations in PLWH on long-term cART and matched HIV-negative controls (HC). The study has used both untargeted and targeted metabolomics to understand the metabolic alterations. The study had also performed in vitro studies to identify metabolic state of the lymphocytic and pro-monocytic HIV-1 latent cell models. 237-238 275-278 I would suggest to be descriptive on how altered glutaminolysis in PLWH is linked to accelerated neurocognitive again and how the hypothesis was deduced. Also it will be a good addition to the discussion if explained the results are relevant to other population.

Other than that overall, the paper has done a comprehensive study in understanding the metabolic changes in PLWH on long-term cART by using system biology methods.

Reviewer #2 (Remarks to the Author):

Comments for Author :

The manuscript entitled "Trans cohorts metabolic reprogramming towards glutaminolysis in long-term successfully treated HIV-Infection: potential role in accelerated aging and latency reversal " is quite interesting and describes the metabolic reprogramming in people living with HIV and the glycolysis and glutaminolysis in HIV-latency cell models.

However, to publish this paper some additional improvement of should be done:

- a) The authors claim that "The only clinical parameter that achieved a statistically significant difference ($p=0.007$) between HC and cART patients was exercise", however, the impact of exercise was not considered in the subsequent analysis. Besides, the duration of treatment is also a very important factor. Author should take these two factors into consideration.
- b) The authors need to provide the clinical and demographic information of the Indian cohorts.
- c) The authors should provide details of untargeted metabolomics analysis. According to the method mentioned in "Plasma Metabolic Signature and Abnormalities in HIV-Infected Individuals on Long-Term Successful Antiretroviral Therapy", authors claimed that "The methanol extract was divided into four fractions: two for analysis by two separate reverse-phase (RP)/UPLC/MS/MS methods with positive ion mode electrospray ionization (ESI), one for analysis by RP/UPLC/MS/MS with negative ion mode ESI, and one for analysis by HILIC/UPLC/MS/MS with negative ion mode ESI." How did the authors integrate these four sets of data?
- d) It is recommended that authors use the targeted method to verify the identification and content changes of these five overlapping biomarkers (5 α -androstane-3 α ,17 β -diol monosulfate, androsterone sulfate, epiandrosterone sulfate, metabolonic lactone sulfate and methionine sulfone).
- e) Did the untargeted metabolomics detection of Cameroon and Indian cohorts conducted separately? If so, how does the author deal with data from different batches.
- f) -page 4; line 97 " ...environmental factors: diet, genetics, microbiome,"
- g) -page 19; the figure legend of Figure 5

Reviewer #3 (Remarks to the Author):

The authors have done a wonderful job analysing and visualising their data. The manuscript is very well written, and I really enjoy reading through it.

Some minor comments below:

- Line 57: Brief description of such alterations could be included in the introduction
- Any possible explanation for the differences in the amino acid profiles between the two cohorts (Line 157)?

- Please include cell culture conditions in your methods
- Please discuss the limitations of your study
- Please include a list of abbreviations, group IDs, legends etc in the figure legends (especially supplementary figures).
- Figure S4: metabolite names are cut in some sub-plots. Please correct.
- Figure S5: add biomarker names/IDs
- Link to IPA software not working
- GitHub link not working

Response to reviewers:

Reviewer #1 (Remarks to the Author):

This study investigates the metabolic alterations in PLWH on long-term cART and matched HIV-negative controls (HC). The study has used both untargeted and targeted metabolomics to understand the metabolic alterations. The study had also performed in vitro studies to identify metabolic state of the lymphocytic and pro-monocytic HIV-1 latent cell models. 237-238 275-278 I would suggest to be descriptive on how altered glutaminolysis in PLWH is linked to accelerated neurocognitive again and how the hypothesis was deduced. Also it will be a good addition to the discussion if explained the results are relevant to other population.

Other than that overall, the paper has done a comprehensive study in understanding the metabolic changes in PLWH on long-term cART by using system biology methods.

Response: We are thankful for the positive response in our study. We already discussed this in the 4th paragraph in the discussion section in the first version of the manuscript (Line 259-277).

Reviewer #2 (Remarks to the Author):

The manuscript entitled "Trans cohorts metabolic reprogramming towards glutaminolysis in long-term successfully treated HIV-Infection: potential role in accelerated aging and latency reversal " is quite interesting and describes the metabolic reprogramming in people living with HIV and the glycolysis and glutaminolysis in HIV-latency cell models. However, to publish this paper some additional improvement of should be done:

a) The authors claim that "The only clinical parameter that achieved a statistically significant difference ($p=0.007$) between HC and cART patients was exercise", however, the impact of exercise was not considered in the subsequent analysis. Besides, the duration of treatment is also a very important factor. Author should take these two factors into consideration.

Response: We are thankful to the reviewer for the comments. We have already presented the analysis as supplementary Fig S4 and stated the same in line 133 as follows "*Furthermore, after correcting for confounders using multivariate linear regression, all metabolites were statistically significant (Fig S4).*".

b) The authors need to provide the clinical and demographic information of the Indian cohorts.

Response: Thank you for the suggestion. The clinical data for the Indian cohort was presented in our earlier paper (Babu et al 2020, Metabolites). We have now added the clinical data for both cohorts as supplementary table S1 on the samples that are used in targeted analysis.

c) The authors should provide details of untargeted metabolomics analysis. According to the method mentioned in "Plasma Metabolic Signature and Abnormalities in HIV-Infected Individuals on Long-Term Successful Antiretroviral Therapy", authors claimed that "The methanol extract was divided into four fractions: two for analysis by two separate reverse-phase (RP)/UPLC/MS/MS methods with positive ion mode electrospray ionization (ESI), one for analysis by RP/UPLC/MS/MS with negative ion mode ESI, and one for analysis by HILIC/UPLC/MS/MS with negative ion mode ESI." How did the authors integrate these four sets of data?

Response: We are thankful to the reviewer. The analysis was performed by Metabolon, Inc as per their proprietary analysis pipeline. The method is ISO 9001:2015 certified. We received the

Response to reviewers:

merged data that we provided as a supplementary file in figshare. We have now mentioned this in the text line 337.

d) It is recommended that authors use the targeted method to verify the identification and content changes of these five overlapping biomarkers (5 α -androstan-3 α ,17 β -diol monosulfate, androsterone sulfate, epiandrosterone sulfate, metabolonic lactone sulfate and methionine sulfone).

Response: We are thankful to the reviewer for this suggestion. Unfortunately, we don't have any samples left. Moreover, no assay kit directly measures those metabolites. As these are HIV-infected samples, all facilities do not accept the samples and we don't have an in-house method available for these.

e) Did the untargeted metabolomics detection of Cameroon and Indian cohorts conducted separately? If so, how does the author deal with data from different batches.

Response: This is a very interesting question, thank you for that. We have not merged the cohorts. As we stated that in targeted analysis "*Even though the samples were run together, there was a clear cohort effect (Fig S6).*" Therefore all the analyses were done in a cohort-specific manner. Despite that fact, our findings were consensus in both cohorts. However we noticed that we have written "To identify common biomarkers associated with HIV status and the impact of cART and strengthen our study, we **combined** the data from our Cameroon cohort with untargeted metabolomics analysis from an Indian cohort." We have now replaced the word combined with compared. Thank you for pointing out this.

f) -page 4; line 97 " ...environmental factors: diet, genetics, microbiome,"

Response: Thank you for pointing out the redundancies. We have now removed the word "environmental factors:" from the text.

g) -page 19; the figure legend of Figure 5

Response: Sorry we didn't understand this. Figure legend is already there "**Figure 5. Effect of cART regimens on HIV activation during inhibition of glutaminolysis:.....**"

Reviewer #3 (Remarks to the Author):

The authors have done a wonderful job analysing and visualising their data. The manuscript is very well written, and I really enjoy reading through it.

Response: We are thankful to the reviewer for positive comment in our study.

Some minor comments below:

- Line 57: Brief description of such alterations could be included in the introduction

Response: Thank you for the suggestion. We have now included more common alterations i.e. **amino acid and fatty acid metabolism**. We already discussed this in the discussion second paragraph. Line 240-251

- Any possible explanation for the differences in the amino acid profiles between the two cohorts (Line 157)?

Response to reviewers:

Response: Thank you for the suggestion. One of the possible explanations is that the individuals in the Cameroon Cohorts are meat-eaters. However we don't have the diet data from the Indian cohort, which restricts us from speculating.

- Please include cell culture conditions in your methods

Response: Thank you for the suggestion. We have now included that in the method section as follows: Line:332-334.

"All the cells were maintained in RPMI-1640 (Sigma, USA) supplemented with 10% fetal bovine serum (Gibco, USA) and 20 units/mL penicillin and 20ug/mL streptomycin (Sigma, USA) under 5% CO₂ and 37°C temperature."

- Please discuss the limitations of your study

Response: Thank you for the suggestion we have now added the limitations of the study as follows:

The study has some limitations that merit comments. First, it is a cross-sectional study so the analysis was restricted to the association study, no causality can be inferred. Second, despite having the patients from two cohorts the number of samples was relatively small. Third, though we corrected multiple hypotheses, the high number of associations may have led to type I errors. Finally, the Cameroon cohort was a clinically highly selected cohort with long-term successful therapy that may not represent the general population. However the biggest strength of our study is the use of two cohorts to validate the findings and identify the commonality between the cohorts associated with successful cART in PLWH.

- Please include a list of abbreviations, group IDs, legends etc in the figure legends (especially supplementary figures).

- Figure S4: metabolite names are cut in some sub-plots. Please correct.

- Figure S5: add biomarker names/IDs

Response: We are sorry for the mistakes. We corrected those.

- Link to IPA software not working

Response: Thank you for checking this. The company has changed the link. As it is commercial software, we now mention the company and country.

- GitHub link not working

Response: The link was correct, it was in the private mode. We have now made it public.

REVIEWERS' COMMENTS:

Reviewer #2 (Remarks to the Author):

one minor correction:

-page 20; the figure legend of Figure 5. Authors use "Figure1:, Figure 2:,Figure 3:, Figure 4:" in the other four figure legends while "Figure 5." in the figure legend of Figure 5.

Reviewer #3 (Remarks to the Author):

The authors have successfully addressed all my comments. I am happy to accept the manuscript for publication as is.

Response to reviewers:

We are thankful to all the reviewers and Editor for their constructive criticism and for accepting our manuscript

Rev#2

one minor correction:

-page 20; the figure legend of Figure 5. Authors use "Figure1:, Figure 2:,Figure 3:, Figure 4:" in the other four figure legends while "Figure 5." in the figure legend of Figure 5.

Response: We have now corrected line 661.